# Inhibition of Histone Deacetylase Activity Increases Cisplatin Efficacy to Eliminate Metastatic Cells in Pediatric Liver Cancers

**DOI:** 10.3390/cancers16132300

**Published:** 2024-06-22

**Authors:** Ruhi Gulati, Yasmeen Fleifil, Katherine Jennings, Alex Bondoc, Greg Tiao, James Geller, Lubov Timchenko, Nikolai Timchenko

**Affiliations:** 1Division of General and Thoracic Surgery, Cincinnati Children’s Hospital Medical Center, Cincinnati, OH 45229, USA; ruhi.gulati@cchmc.org (R.G.); yasmeen.fleifil@cchmc.org (Y.F.); alex.bondoc@cchmc.org (A.B.); greg.tiao@cchmc.org (G.T.); 2Department of Neurology, Cincinnati Children’s Hospital Medical Center, Cincinnati, OH 45229, USA; katherine.jennings@cchmc.org (K.J.); lubov.timchenko@cchmc.org (L.T.); 3Department of Surgery, University of Cincinnati College of Medicine, Cincinnati, OH 45229, USA; 4Division of Oncology, Cincinnati Children’s Hospital Medical Center, Cincinnati, OH 45229, USA; james.geller@cchmc.org

**Keywords:** hepatoblastoma, hepatocellular carcinoma, metastases, HDAC, Sp5, cisplatin

## Abstract

**Simple Summary:**

Patients with pediatric liver cancers hepatoblastoma and hepatocellular carcinoma very often develop lung metastases. These cancers can present with lung metastases and are at higher risk of relapse. Although cisplatin is very effective at clearing lung metastases, they can still relapse. Therefore, there is an urgent need to develop therapeutic approaches to prevent the development of lung metastases in patients with pediatric liver cancers. In this paper, we show that the metastatic microenvironment of HBL and HCC patients contains a heterogeneous cell population that formed tumor clusters. We found that both fresh primary tumors and generated primary cell cultures had increased the expression of HDAC1, a histone deacetylase, and the transcription factor Sp5. Sp5 and HDAC1 work in tandem by transporting HDAC1 to the promoters of genes and changing their expression. We analyzed the effects of the HDAC inhibitor, SAHA, on the metastasis-initiating cells in combination with cisplatin. We found that HDAC inhibition increases the efficacy of cisplatin to eliminate these metastasis-initiating cells.

**Abstract:**

The pediatric liver cancers, hepatoblastoma and hepatocellular carcinoma, are dangerous cancers which often spread to the lungs. Although treatments with cisplatin significantly improve outcomes, cisplatin may not eliminate metastasis-initiating cells. Our group has recently shown that the metastatic microenvironments of hepatoblastoma contain Cancer Associated Fibroblasts (CAFs) and neuron-like cells, which initiate cancer spread from liver to lungs. In this study, we found that these cells express high levels of HDAC1; therefore, we examined if histone deacetylase inhibition improves cisplatin anti-proliferative effects and reduces the formation of tumor clusters in pediatric liver cancer metastatic microenvironments. Methods: New cell lines were generated from primary hepatoblastoma liver tumors (hbl) and lung metastases (LM) of HBL patients. In addition, cell lines were generated from hepatocellular neoplasm, not otherwise specified (HCN-NOS) tumor samples, and hcc cell lines. Hbl, LM and hcc cells were treated with cisplatin, SAHA or in combination. The effect of these drugs on the number of cells, formation of tumor clusters and HDAC1-Sp5-p21 axis were examined. Results: Both HBL and HCC tissue specimens have increased HDAC1-Sp5 pathway activation, recapitulated in cell lines generated from the tumors. HDAC inhibition with vorinostat (SAHA) increases cisplatin efficacy to eliminate CAFs in hbl and in hcc cell lines. Although the neuron-like cells survive the combined treatments, proliferation was inhibited. Notably, combining SAHA with cisplatin overcame cisplatin resistance in an LM cell line from an aggressive case with multiple metastases. Underlying mechanisms of this enhanced inhibition include suppression of the HDAC1-Sp5 pathway and elevation of an inhibitor of proliferation *p21*. Similar findings were found with gemcitabine treatments suggesting that elimination of proliferative CAFs cells is a key event in the inhibition of mitotic microenvironment. Conclusions: Our studies demonstrate the synergistic benefits of HDAC inhibition and cisplatin to eliminate metastasis-initiating cells in pediatric liver cancers.

## 1. Introduction

Pediatric liver cancers Hepatoblastoma (HBL), Hepatocellular carcinoma (HCC) and Hepatocellular carcinoma not otherwise specified (HCN-NOS) require aggressive surgical and chemotherapeutic treatment, and at times prove lethal [1,2]. In contrast to adult liver cancers, pediatric cancers develop at a very young age when livers have limited nutritional and environmental stressors [1,2,3]. Outside of uncommon predisposing conditions such as Familial Polyposis Coli or Beckwith–Wiedemann Syndrome, genetic analyses of pediatric liver tumors show only rare somatic mutations, with the most common being *β-catenin*, *NRF2* and *TERT* [3,4]. Therefore, it is highly likely that mutation-independent pathways are involved in the development of pediatric liver cancers. In this regard, many recent reports provided evidence for the role of epigenetic alterations of gene expression in pediatric liver cancers. To that end, we have previously shown that post-translational modifications of tumor suppressor proteins convert tumor suppressors C/EBPα, p53 and CUGBP1 into proteins with oncogenic activities [5,6,7,8]. Additionally, β-catenin activation by phosphorylation at Ser675 and subsequent oncogene activation via openings in specific chromosomal regions such as Cancer Enhancing Genomic regions or Aggressive Liver Cancer Domains (CEGRs/ALCDs) drive pediatric liver tumor oncogenesis [9,10,11,12,13]. These pathways play an essential role specifically in the development of HBL and fibrolamellar hepatocellular carcinoma [11,12,13].

Recent studies of HBL have identified epigenetic changes in the levels of DNA methylation, microRNAs, LncRNAs and histone modification [14]. Rivas and colleagues first found that DNA methylation is an essential epigenetic player in HBL [15]. Following studies from this group have analyzed the expression of 24 genes associated with DNA methylation in a large biobank of HBL specimens and found a significant decrease in DNA methylation enzymes [16]. In agreement with these reports, recent studies revealed that the inhibition of methyltransferase activity of G9a/DNMT1 reduces the proliferation of HBL cells and growth of HBL spheroids [17]. The critical role of microRNAs, as well as the role of LncRNAs in the development of HBL, was also recently described in several papers [14,18,19]. An additional epigenetic pathway in hepatoblastoma and in HCC is related to the modification of m6A in mRNAs [20,21].

Among these epigenetic pathways, histone modifications have been much more investigated and seem to make an essential contribution to the development of pediatric cancers, including methylation, acetylation, and deacetylation on the K residues of histones. The main chromatin remodeling proteins that acetylate/deacetylate histones and change the chromatin structure are histone acetylase p300 and a family of histone deacetylases (HDAC). Early studies by Kappler’s group showed that *HDAC1* and *HDAC2* are elevated in HBL, and that the HDAC inhibition by SAHA and MC1568 resulted in a reduction in cell viability, apoptosis induction, reactivation of tumor suppressor genes, and inhibition of proliferation in HBL cell lines [22]. HDAC proteins do not bind to DNA directly and are usually delivered to chromatin/DNA by transcription factors such as Sp5 [10]. In this regard, examinations of precise molecular mechanisms of HDAC-dependent epigenetic alterations in HBL patients revealed that *HDAC1* and transcription factor *Sp5* are elevated in HBL, form HDAC1-Sp5 complexes, and repress the expression of hepatocyte markers and inhibitor of proliferation *p21* [10]. In agreement, it has been shown that inhibited HDAC activity in patient-derived xenograft (PDX) models inhibits the growth of implanted tumors [23]. Despite progress being made in the study of HDAC activity within pediatric liver cancers, little is known about the metastatic cancer cells that are potentially driven by HDAC signaling. Precise mechanisms by which HDAC inhibition suppresses tumor growth are also unknown. This manuscript describes a study of the HDAC1-dependent liver cancer pathways in a metastatic microenvironment of pediatric liver cancers. We present evidence that HDAC inhibition increases cisplatin efficacy in eliminating cancer cells via the repression of HDAC1-Sp5, and the subsequent de-repression of proliferation inhibitor *p21* manifests as inhibition of tumor clusters.

## 2. Materials and Methods

Pediatric HBL patients and tumor models: This study was approved by the Institutional Review Board (IRB) at CCHMC (protocol number 2016-9497). Informed consent was obtained from each study patient, or their parents as indicated, prior to obtaining specimens. In this study, we have investigated specimens from a biorepository of specimens from 46 HBL patients, 3 HCC patients, 1 HCN-NOS (#77) and 2 specimens of lung metastases from HBL patients. The liver tumor #31 and two lung metastases #60 and #81 were obtained from the same patient after repetitive developments of lung metastases and subsequent surgeries. Seven other HBL patients in this study were diagnosed with lung metastases before or after primary surgery. Background regions include the sections of “healthy” portions of liver of the same patients that are adjacent to the tumor section whereas tumor sections are labelled as “hepatoblastoma (HBL)”. The age of patients was from 1 to 3 years. Note that, in many cases, we have received limited amounts of the specimens that were sufficient only for RNA analysis and for immunostaining. The studies described in this manuscript used 16 hbl, 3 hcc, 1 hcn-nos and 2 LM cell lines

Examination of HepG2 cells: HepG2 cells were maintained in Dulbecco’s Modified Eagle Medium (DMEM, Fisher Scientific 11-965-092) supplemented with 10% FBS and penicillin/streptomycin in a 37 °C, 5% CO_2_ incubator as described in our paper [13]. Cells were plated at either a low density or high density. Immunohistochemistry for NeuN was performed as described previously [11,12,13]. HepG2 cells were treated with cisplatin (1 µg/mL) and images were taken at 48 after initiation of treatments. Protein extracts were isolated from experimental plates and used for Western Blot with antibodies to HDAC1, p21, NeuN, β-III-tubulin, and markers of CAFs.

Cell-free-exit protocol for generation of cells with metastatic activities: We developed a methodology for isolation of cells from liver tumors with potential metastatic activities (metastasis-initiating cells) [13]. Small fragments of tumor were plated on collagen plates containing DMEM with 10% FBS. Released cells were monitored for several weeks. Several examples of cell exit from tumors are shown in the manuscript. Experiments with primary cell lines were performed with cells from passages 2–4, a time during which the natural heterogeneous metastatic microenvironments are preserved, with the tumor tissue exiting cells having an approximate 1:1 ratio of CAFs to neuron-like cells. Since CAFs grow faster than neuron-like cells, at later passages, the balance is shifted to CAFs. For clarity of presentation, we use capital HBL and HCC for patients’ tissue samples and lower case hbl and hcc/hcn-nos for generated primary cultures. We noted that all examined cell lines did not become senescent at these early passages. Future work will show if these cells might be immortalized and used for long term studies.

Treatments of cells with cisplatin, SAHA and with a combination of cisplatin and SAHA: Two methods were applied for the treatments of primary cell cultures. The first method included plating cells at low-density (approximately 10–15%) and monitoring their growth and formation of tumor clusters for 10–20 days. Under normal conditions, the cells proliferate, interact with each other, and form tumor clusters [13]. The cells were treated with DMSO, 1 μg/mL cisplatin, 1 μM SAHA, or a combination of 1 μg/mL cisplatin and 1 μM SAHA (cis+SAHA). Growth and cluster formation were monitored, and images were taken on days 10–12 and 20. The number of tumor clusters was counted in each treatment. Whole-cell protein extracts were isolated from the plates and used for Western Blot and Co-IP studies. Tumor clusters are difficult to trypsinize, making calculations in the low-density protocol difficult. Thus, we added the high-density protocol, in which cells did not make clusters but proliferated quickly within the timespan for experiments. This high-density plating protocol included plating cells approximately at 35–40% density, and treatments with cisplatin, SAHA, and cis+SAHA. A subsequent analysis of the outcome was performed 48 h after treatment. The number of cells was counted, and protein expression was analyzed by Western Blot. The cisplatin-resistant cells are named if (1) cisplatin did not block formation of clusters in the low-density plating protocol; and (2) if cisplatin did not reduce number of cells in the high-density plating protocol. 

Antibodies: HDAC1 (EMD Millipore Corp, clone 2E10), Sp5 (Abcam, ab36593), β-catenin (Abcam, [E247] ab32572); ph-S675-β-catenin (Cell Signaling Tech, 4176S); TCF4 (Cell Signaling, C48H11), p300 (Invitrogen, PA1-848), p21 (Santa Cruz, sc-6246), β-III tubulin (Abcam, ab18207), NeuN (Abcam, ab104225), NGF (Abcam, ab52918) and α-SMA (Cell Signaling,14968). The dilutions of antibodies in Western Blot were adjusted for each type of antibodies. The range of dilutions was from 1/5000 (β-actin) to 1/200 (p300 and p21).

TaqMan Probes: CTNNB1 (Hs00355045_m1), TUBB3 (Hs00801390_s1), RBFOX3 (Hs01370653_m1), NGF (Hs00171458_m1), ACTA2 (Hs00426835_g1), Col1a1 (Hs00164004_m1), Col1a2 (Hs01028956_m1), HDAC1 (Hs02621185_s1), HDAC2 (Hs00231032_m1), HDAC3 (Hs00187320_m1), HDAC4 (Hs01041648_m1), SIRT1 (Hs01009006_m1), HDAC11 (Hs00978031_g1) and 18s (Hs03003631_g1).

Real-Time Quantitative Reverse Transcriptase-PCR: RNA was isolated by the TRIzol/chloroform extraction method and used for cDNA synthesis. cDNA was synthesized using 2 μg of RNA applying the High-Capacity RNA-to-cDNA kit (Applied Biosystems, 4387406). QRT-PCR was performed using TaqMan probes and TaqMan Gene Expression Master Mix (Applied Biosystems, 4369016) and analyzed using the delta-delta CT method. mRNA levels were quantified using GraphPad Prism 9.5 software.

Protein Isolation, Western Blotting, Co-Immunoprecipitation (Co-IPs): Whole Cell Extracts (WCEs) were isolated and Western Blot was performed as described previously [10,11,12,13]. 30–50 μg of proteins was loaded on the gels. Appendix A show whole gel images of Western Blots. To increase the sensitivity of the Co-IP approach, an improved True Blot protocol was used as previously described [8,9]. Protein levels were quantitated as ratios to β-actin using ImageJ software.

Statistical Analysis: All continuous values are presented as mean + SEM using GraphPad Prism 9.5. Where indicated, Student’s *t*-tests and One-Way ANOVA analyses were used. *p* value < 0.05 was considered significant.

## 3. Results

Commonly used hepatoblastoma cell line HepG2 contains HDAC positive neuron-like cells that are resistant to cisplatin treatment. Our group recently found that the natural metastatic microenvironment of HBL tumors contains neuron-like cells that are positive for neuronal markers β-III-tubulin and NeuN [13]. Since our previous studies showed that neuron-like cells grow faster under conditions of low-density plating [13], we examined the cell shape and transcriptomic signature of HepG2 cells after low-density and high-density plating. Figure 1A displays that a week after low-density plating, the HepG2 line contains cells with the shape of neurons interacting with other cells. The HepG2 cells plated at high density have rare but distinct cells with the shape of neurons (Figure 1A, right). It is interesting that, on low-density plates, we observed mainly 5–7 cells with the shape of neurons. Identical numbers of neuron-like cells have been seen on the plates with high density plating. On these plates, however, most cells formed large clusters of cells (Figure 1A). Immunostaining with β-III-tubulin confirmed that the HepG2 line contains neuron-like cells (Figure 1B). We next examined the sensitivity of neuron-like cells in the HepG2 cell line to cisplatin. Figure 1C reveals that HepG2 cells with the shape of neurons are resistant to cisplatin. This result agrees with a previous study that detected chemo-resistant HepG2 cells with the shape of neurons which are positive for a neuronal stem marker CD133 [24]. Western blot analyses with β-III-tubulin and NeuN confirmed that the cells resistant to cisplatin treatments are positive for markers of neurons (Figure 1D). Since our previous studies unveiled the critical role of HDAC1 in the promotion of pediatric liver cancer [10], we examined HDAC1 and its downstream target p21 in cisplatin treated HepG2 cells. We found that HDAC1 is not detectable in cisplatin-resistant cells, while p21 was increased (Figure 1D). To examine if apoptosis is increased in cisplatin-treated cells, Western Blots for PARP1 were conducted and found that the 89kD PARP1 cleavage fragment is abundant in cisplatin-treated HepG2 cells, revealing apoptosis.

We next analyzed the HDAC1-Sp5 pathway and neuronal pathways in a fresh biobank of pediatric liver cancer samples, including HBL, HCC, and HCN-NOS (*n* = 36). In addition, we examined hepatocyte markers (targets of HDAC1), stem cell markers, CAF markers, as well as targets of the β-catenin-CEGRs/ALCDs axis [7,11,12,13]. Figure 1E shows that *HDAC1*, *Sp5*, and neuronal markers are increased in many of the pediatric liver tumors studied, while the hepatocyte markers, which are repressed by the HDAC1-Sp5 axis, are reduced. We also found that *β-catenin*, CAF markers, stem cell markers, and β-catenin-CEGRs/ALCDs dependent oncogenes are increased within liver tumors of about 70% of HBL patients in our biobank. HDAC1 is just one part of the HDAC family. Since this study utilizes SAHA, an HDAC inhibitor, it was crucial that we analyze all members within this target family, including *HDAC2*, *HDAC3*, *HDAC4*, *HDAC11*, and *SIRT1*, within our HBL biobank. Figure 1F demonstrates that *HDAC2*, *HDAC3*, *HDAC4*, *HDAC11,* and *SIRT1* are elevated in the majority of HBL tumors.

Characterization of liver cancer pathways in tumor samples: We generated cell lines from HBL, HCC, HCN-NOS, and lung metastases patient tumors, which were resistant to cisplatin treatments and demonstrated elevated HDAC1 and Sp5 proteins. Figure 2A shows the list of samples that were used for the generation of cultured cell lines. In Figure 2B, we examined the HDAC1-Sp5 pathway within the initial tumors that were used for cell line generation. QRT-PCR showed that *HDAC1* and *Sp5* are increased in 18 out of 21 tumors used for the generation of cell lines. Examination of hepatocyte markers displayed a strong reduction, correlating with the elevation of HDAC1-Sp5 in each HBL sample (Figure 2C). Interestingly, stem cell markers and neuronal markers are also increased in patients’ tumors with an elevated HDAC1-Sp5 pathway (Figure 2D,E). Thus, transcriptome profiling of tissue from HBL, HCC, HCN-NOS, and lung metastases samples revealed that the HDAC1-Sp5 axis is elevated and downstream targets are reduced, while markers of liver aggressiveness are elevated in these patients. To confirm the increase in HDAC1 in HBL samples, immunohistochemistry was performed. The HBL tumors revealed stronger HDAC1 staining than background (adjacent to tumor) regions. The increased HDAC1 signals were observed in both hepatocytes and in tumor areas with fiber-like structures (Figure 2F). Since our previous work showed that hepatocytes do not exit the tumors in a “free exit” protocol [13], we suggest that the HDAC1-positive fibers release HDAC1-positive cells.

Generation of patient derived cell lines with metastatic activities. Given the elevation of the HDAC1-Sp5 pathway seen in patients with pediatric liver cancers (Figure 2), we have utilized a recently established protocol for generating cell lines with metastatic activities [13] and have established new cell lines from these patients. Metastatic cells must have the ability to exit from the tumor (free exit), the ability to proliferate, interact with each other and form tumor clusters. Our cell free exit protocol is designed to generate these cells (Figure 3A). Immediately after surgery, a piece of resected tumor is placed on a collagen plate. Within a short time, the tumor is usually attached to the plate and starts a release of the metastatic cells in media, where cells become attached to the plate and start proliferation. Once cells reached high density, they were trypsinized and sub-cultured on fresh plates for further growth and analyses. Figure 3B shows an example of the cell exit from the tumors. Three days post-attachment, the tumors released few cells that are observed near the tumor. In one week, more cells are released, further away from the tumor. After two weeks and beyond, we found cases where tumors released clusters of cells (Figure 3B, right). Further studies with generated cell lines revealed that, when the cells are plated at low density, the cells proliferate, display intensive cell–cell interactions (Figure 3C) and form tumor clusters with a well-organized center (Figure 3D). We have observed that the cells released by both primary tumors and lung metastases contain Cancer Associated Fibroblasts (CAFs) and neuron-like cells [13]. Consistent with this, immunostaining of the new cell lines with NeuN revealed that a large portion of the released cells are neuron-like cells, and that they make up most of the clusters at two weeks post-plating (Figure 3E). Thus, these studies have demonstrated that the new patient-derived cell lines display metastatic activities—including free exit from the tumor, proliferation, cell–cell interactions and formation of tumor clusters. Further, we examined the formation of tumor clusters as an outcome of drugs treatments in untreated and treated cells.

Generated hbl cell lines express high levels of HDAC1 and form HDAC1-positive tumor clusters. We next asked if the generated hbl cell lines preserved the HDAC1-Sp5 pathway. Western blot analysis of proteins from original HBL tumors and from the hbl cell lines showed that HDAC1 and Sp5 are increased in hbl cells, while p21 is reduced (Figure 4A). We also found that hbl107, hbl109, and hbl111 cells are positive for HDAC1 and that they form HDAC1-positive tumor clusters (Figure 4B).

HDAC inhibition in low-density plated hbl cells suppresses the formation of tumor clusters and enhances cisplatin efficacy to reduce proliferation. In our studies, the majority of generated hbl cell lines were derived from patients who had already undergone treatment with cisplatin, suggesting that the cells exiting tumor tissue during cell line generation are possibly resistant to cisplatin as part of a selection process. Some of these patients had already developed lung metastases at the time of diagnosis or after the primary tumor resection. Given the high levels of HDAC1 in these cells and our hypotheses that epigenetic regulation is key in driving pediatric liver tumors, we asked if HDAC inhibition might enhance cisplatin efficacy to inhibit the proliferation of hbl cells and reduce tumor cluster formation. We have previously found that the metastatic associated activities of hbl cells include the formation of tumor clusters if the cells are plated at low density [13]. Therefore, we have applied this protocol to test if HDAC inhibition suppresses the formation of tumor clusters which might improve cisplatin-based therapy. The hbl cell lines (hbl69, hbl75, hbl92, and hbl111) were plated at low density and treated with DMSO, cisplatin (1 μg/mL), SAHA (1 μM) and cis+SAHA. Results of the examination of tumor clusters at 12 days after plating are shown in Figure 4C,D. The hbl cells treated with DMSO have formed 8–14 tumor clusters per plate. Although treatments with cisplatin reduced the formation of clusters, 4–6 clusters per plate were detected. Treatments with SAHA alone strongly reduced the formation of big tumor clusters, but the small clusters were still detectable. On the contrary, cis+SAHA treatments completely blocked the formation of tumor clusters and reduced the number of cells. Interestingly, after combined treatments, most surviving cells were single and had the shape of neurons (Figure 4C).

The protein expression of HDAC1, p21, α-SMA (a marker of CAFs) and NeuN in the treated hbl69 and hbl75 cells was examined (Figure 4E). As shown, cisplatin does not change the expression of these proteins; however, treatments with SAHA reduced HDAC1 and α-SMA, and increased the expression of p21 and NeuN (Figure 4E). Combined treatments found a stronger reduction in HDAC1 and α-SMA, and a higher increase in p21. Given that α-SMA is a marker of CAFs and NeuN is a neuronal marker, this data suggest that the combined treatments eliminate CAFs, while neuron-positive cells are resistant and are expressing high levels of p21. When doubling the experimental conditions to 20 days, the number of tumor clusters had increased in DMSO-treated cells to 20–23 per plate. The number of clusters in cisplatin-treated cells was 15–18 clusters per plate (Figure 4F). The size of the tumor clusters was also increased in DMSO-treated cells. In addition, we found that the longer time cell growth after low-density plating leads to physical interactions of the clusters with each other. On the contrary, cells treated with SAHA and with cis+SAHA did not form clusters 20 days after plating. Thus, the cisplatin and SAHA treatments of hbl cells plated at low density showed that cisplatin alone slightly reduces the formation of tumor clusters, while SAHA had anti-tumor activity with a reduction in large tumor clusters, and cis+SAHA completely blocked the development of tumor clusters.

HDAC inhibition in hbl cells plated at high density increases the cisplatin efficacy in eliminating cancer cells. The treatment studies described above were performed with the protocol which includes low density plating and subsequent proliferation of cells that form tumor clusters at two weeks after plating. Although the low-density studies provided significant information, the plating protocol limitations relate to difficulties in counting the number of individual cells within the clusters. The hbl cells plated at approximately 35% density reach about 80% confluence 48 h after plating, usually without the formation of tumor clusters. Four hbl cell lines, including hbl75, hbl74, hbl92 and hbl107, were grown using this protocol and treated with cisplatin, SAHA and a combination. Results of the examination of hbl74, hbl75 and hbl92 cells are shown in Figure 5A. DMSO-treated cells proliferated well and reached 70–80% confluence in 48 h after the treatment. We found that cisplatin alone does not significantly reduce the number of cells. However, combined treatment reduces the number of hbl cells by about 80% (Figure 5A,B). Thus, cis+SAHA might have synergistic effects on hbl cell growth.

The HDAC1-p21 pathway in the treated hbl cells was examined by Western blot analysis. α-SMA and NeuN were also examined. As shown, cisplatin alone does not change HDAC1, p21, α-SMA and NeuN levels; however, treatments with SAHA with or without cisplatin inhibited the expression of HDAC1 and increased HDAC1 target p21 (Figure 5C,D). α-SMA was undetectable by Western blot analysis in hbl cells treated with cis+SAHA. In contrast, NeuN was low or undetectable in hbl cells treated with DMSO, cisplatin or SAHA, but it was increased in hbl cells treated with cis+SAHA. This pattern of protein expression is consistent with the finding that CAFs were almost eliminated by combinatory treatments in hbl cells; but neuron-like cells are resistant to these treatments. (Figure 5E). Since neuron-like cells have elevated p21, we suggest that neuron-like cells are growth arrested on the cis-SAHA treated plates. Similar results were also observed in hbl107 cells (Appendix A).

Generation of cell lines from pediatric patients with HCC. Using the cell free exit protocol (Figure 3A), we have also generated hcc cell lines from patients with HCC or HCN-NOS liver cancers, HCN-NOS77, HCC79, HCC84 and HCC105. Examination of cancer-specific pathways in the HCC tumors showed that HDAC1-Sp5 pathway is increased, while hepatocyte markers are reduced (Figure 6A). Stem cell markers, CEGRs/ALCDs dependent oncogenes, neuronal markers, fibrosis markers and fatty liver markers are also elevated in half of these patients (Figure 6A). Immunostaining of HCC84, HCN-NOS77 and HCC105 tumors to HDAC1 revealed HDAC1-positive cells/fibers (Figure 6B,C). Since neuronal markers are highly expressed in the tumors of pediatric HCC patients, we examined NeuN expression in HCC through immunohistochemistry. A typical example of these studies using HCN-NOS77 is shown in Figure 6D. As shown, NeuN signals are much stronger in HCN-NOS77 tumor sections compared to non-tumor adjacent regions. Brightfield microscopy revealed that generated hcn-nos77 line contains cells with a typical shape of neurons displaying cell–cell interactions (Figure 6E). We found that alterations of the HDAC1-Sp5-p21 pathway are preserved in generated hcc/hcn-nos cell lines (Figure 6F). Western blot-Immunoprecipitation (IP) analysis demonstrated that HDAC1-Sp5 complexes are abundant in hcc/hcn-nos cells (Figure 6F, bottom). Thus, generated pediatric hcc/hcn-nos cell lines reproduce the main characteristics of the pediatric HCC tumors, including activation of the HDAC1-Sp5-p21 pathway, and mirror previously presented findings from HBL studies.

HDAC inhibition in hcc/hcn-nos cell lines enhances the cisplatin efficacy in eliminating cancer cells. Since the HDAC1-Sp5 pathway is activated in hcc/hcn-nos cells, we examined if cells treated with SAHA would increase the effect of cisplatin on hcc/hcn-nos cell proliferation. It was found that hcc/hcn-nos cells grow very slowly; therefore, the cell lines hcn-nos77, hcc79 and hcc105 cells were plated at high density. The response to cisplatin and SAHA in hcc/hcn-nos cells varied. Cisplatin treatments of hcn-nos77 and hcc105 cells resulted in the inhibition of 50% of cells compared to DMSO treatments (Figure 7A). However, hcc79 cells were resistant to cisplatin. Note that post-treatment, the number of cells in cisplatin-treated plates was approximately the same as on the plates before treatment (plating), suggesting that cisplatin inhibits hcc cell proliferation but does not cause cell death. We found that, in all cases, the combined treatments of cis+SAHA caused strong cell reduction (Figure 7B). Examination of HDAC1-p21-α-SMA/neuron-like pathway confirmed that the combined cisplatin and SAHA treatments completely inhibit the expression of HDAC1 and α-SMA, while p21, β-III-tubulin and NeuN are increased (Figure 7C).

Treatments with SAHA increase cisplatin efficacy to eliminate cancer cells generated from lung metastases of HBL patients. Two cell lines from lung metastases HBL patients were generated and characterized [13]. One of these lines was generated from an HBL patient who had multiple surgeries to remove a primary liver tumor (sample #31) and five subsequent lung metastases surgeries, including two last lung metastases (samples LM60 and LM81). We used LM60 and LM81 and a corresponding primary tumor, when available, to examine the expression of markers of hepatocytes, oncogenes and neuronal markers. QRT-PCR showed that most of the hepatocyte markers are dramatically reduced in the primary liver tumor and in lung metastases, while some stem cell markers, β-catenin, neuronal markers and the HDAC1-Sp5 pathway were strongly elevated in both the liver tumor and in lung metastases (Figure 8A). Interestingly, an over 100-fold elevation of Sp5 was observed in the multiple recurrent lung metastasis compared to the background region. Examination of the cells exiting from LM81 revealed that they resemble the shape of neurons, as well as CAFs and are positive for *β-III-tubulin* and *α-SMA* (Figure 8B). Consistent with QRT-PCR results, Western blots and Co-IPs revealed that the HDAC1-Sp5 pathway is elevated in the generated LM81 cell line (Figure 8B, right). Thus, activation of the HDAC1-Sp5 pathway, and elevation of the oncogenic expression observed in the original liver tumor of HBL patients is also preserved in LM81 cells generated from the fifth (multiple recurrent) lung metastasis. 

We next examined the effects of cisplatin and SAHA on the proliferation of LM81 cells and on their ability to form tumor clusters. Like hbl cells, LM cells form tumor clusters under low-density conditions. Therefore, we have applied both low and high-density plating for the treatment of LM81 cells. We found that in DMSO-treated cells plated at low-density, LM81 cells form 14–17 tumor clusters per plate. Cisplatin reduces the number of clusters to 7–10; however, the clusters were still detectable. SAHA alone and in combination with cisplatin blocked the formation of clusters (Figure 8C). LM81 cells plated at high density proliferate quickly, reaching 80–90% confluence within 48 h after plating in both DMSO-treated and cisplatin-treated plates. Cisplatin treatment has no significant effect on the number of cells compared to DMSO-treated LM81 cells. However, SAHA and both cisplatin and SAHA significantly reduced the number of LM81 cells (Figure 8C,D). Interestingly, cis+SAHA has a greater reduction in cells compared with SAHA alone. The reduction in LM81 cells treated with SAHA or cis+SAHA was accompanied by the reduction in HDAC1-Sp5 pathways, inhibition of HDAC1-Sp5 complexes and an increase in p21 (Figure 8E). Examination of CAFs (α-SMA) and neuronal (NGF and β-III-tubulin) markers suggests that CAFs are eliminated in LM81 cells by the combined treatments and that the remaining cells are positive for β-III-tubulin and NGF (Figure 8F).

The inhibition of DNA replication by anti-cancer drug gemcitabine parallels the inhibition of CAFs and blocks the formation of tumor clusters in cells derived from lung metastases. Gemcitabine is an FDA-approved inhibitor of DNA replication and effectively kills cancer cells by increasing apoptosis [24,25,26]. We found that in a low-density setting, LM81 cells treated with gemcitabine (5 nM) do not form tumor clusters, compared to DMSO-treated cells (Figure 9A). In a high-density setting, gemcitabine also significantly reduced the number of LM81 cells (Figure 9A,B). Most cells remaining on the plates after gemcitabine treatments had neuronal features, like that seen with the cis-SAHA treatments. The HDAC1-Sp5 pathway was examined and found that, while HDAC1 is not detectable in gemcitabine-treated cells, p21 was elevated. It was also found that gemcitabine eliminates CAFs, while neuron-like cells are not eliminated (Figure 9C). Thus, gemcitabine strongly inhibits the formation of tumor clusters and proliferation of cells from cultured lung metastasis, perhaps via the elimination of HDAC1-dependent CAFs (Figure 9D), which have a high rate of proliferation.

## 4. Discussion

Adult liver cancer, hepatocellular carcinoma, predominantly develops in aged individuals and is often the result of unbalanced nutrition or other liver disorders. However, pediatric liver cancers occur at a very young age when livers have not been challenged by improper nutrition uptake or other liver disorders. Only a small number of patients with pediatric cancer have germline or genetic mutations, mainly in APC or β-catenin, respectively [27,28,29], suggesting that mutation-independent mechanisms are involved in the development of HBL, HCN-NOS and perhaps pediatric HCC as well. Our group has been investigating such mechanisms, focusing on post-translational modifications of tumor suppressors and on epigenetic upregulation of oncogenes and cancer-related genes [8,9,10,11,12,13]. We have found that two epigenetic pathways are activated in patients with pediatric cancers. The first pathway includes activation of oncogenes via the ph-S675-β-catenin-CEGRs/ALCDs axis [10,11,12,13], while the second epigenetic pathway includes the elevation of *HDAC1* and *Sp5*, along with the subsequent inhibition of hepatocyte markers and the tumor suppressor protein p21 (HDAC1-Sp5-p21 axis) [10]). In this paper, the HDAC1-Sp5-p21 pathway was investigated to understand its involvement in the generation of metastasis-initiating microenvironments of pediatric liver cancers, and if the inhibition of this pathway together with cisplatin is effective in the elimination of cancer cells. Although cisplatin is commonly used for HBL patients, this drug is associated with severe side effects including ototoxicity, neurotoxicity and gastrointestinal and hematologic toxicity [30,31]. Further, some patients with pediatric liver cancers are not adequately treated with cisplatin, which manifests as disease progression, relapse and, at times, tumor-associated mortality. Therefore, we thought to examine if other drugs could be used to replace cisplatin or a combination of cisplatin with other drugs could have a synergistic effect to improve the reduction in cancer cells in HBL and HCC. We found that some HBL and HCC tumor samples have high levels of HDAC1 and Sp5, suggesting that the HDAC inhibitor SAHA could be used to reduce the HDAC1-Sp5 pathway in HBL and HCC. The presented experimental data also demonstrate that HBL, HCC and lung metastases release CAFs and neuron-like cells that have high levels of HDAC1 and Sp5 proteins, as well as low levels of p21. These results suggest that in patients cells under certain conditions are released in the bloodstream, creating a high risk for the development of lung metastases. The limited activity of cisplatin monotherapy in these models may relate to the fact that the tumor models originated from patients previously treated with cisplatin and are therefore subject to evolution of resistance.

Importantly, in all tested cell models with elevated HDAC1, the combinations of treatments with cisplatin and with the HDAC inhibitor SAHA eliminate CAFs, while neuron-like cells survived the combined treatments but did not proliferate. SAHA inhibits the activity of several members of the HDAC family, and as such the inhibition of additional members may be involved in the elimination of CAFs. Interestingly, the HDAC inhibition by SAHA alone in some cell lines had a stronger effect, than cisplatin alone suggesting that HDAC activity is relevant in models derived from patients previously exposed to cisplatin. These findings are consistent with previous studies showing that the HDAC inhibition by SAHA in HepG2 cells increased cisplatin’s ability to eliminate cancer cells [22]. Further, Espinoza et al. have recently reported that the inhibited HDAC activity by panobinostat suppresses tumor growth in patient-derived spheroids and in PDX models [23]. Panobinostat likely utilizes similar molecular mechanisms, including the inhibition of HDAC1-Sp5 and the elevation of p21.

The presented data show the investigation of cell lines generated from an HBL patient with a primary tumor and multiple subsequent lung metastases. This unique case allowed a close examination of key events that drives ongoing recurrences, including a key investigation of a cell line generated from the fifth lung metastasis from this patient. Elevation of the HDAC1-Sp5 pathways was preserved from the primary liver tumor evolving through each lung metastasis. The lung metastases of this patient also preserved increased levels of stem cell markers, oncogenes that are under the control of the β-catenin-CEGRs/ALCDs pathway [13] and neuronal markers. The results of these studies bring us to a critical point which is presented in Figure 10. By generating primary cell lines from fresh tumors, the metastatic microenvironment of pediatric patients is full of potential issues, such as CAFs and neuron-like cells. While treatments with cisplatin alone are not sufficient to inhibit the formation of tumor clusters, a combination of cisplatin with SAHA inhibits the formation of metastatic tumor clusters via the elimination of CAFs. These studies provide crucial translational and mechanistic insights into the role of combined cisplatin/HDAC therapy in reducing tumor growth. Novel therapies that take advantage of the HDAC1 pathway, perhaps in combination with cisplatin or alternative therapies such as gemcitabine, may help to eliminate malignant clones in the metastatic niche and thereby enhance a cure.

## 5. Conclusions

Our work shows that the metastatic microenvironment of pediatric liver cancers consists of CAFs and neuron-like cells which are under control of HDAC1-Sp5 axis. Treatment of this microenvironment with cisplatin alone is not sufficient to eliminate metastatic cells and block development of tumor clusters. However, treatments of the metastatic microenvironment using a combination of cisplatin and the inhibitor of HDAC SAHA blocks the formation of tumor clusters. These findings provide a rationale to evaluate if this combination might work with pediatric HBL and HCC patients in clinical settings.

## Figures and Tables

**Figure 1 cancers-16-02300-f001:**
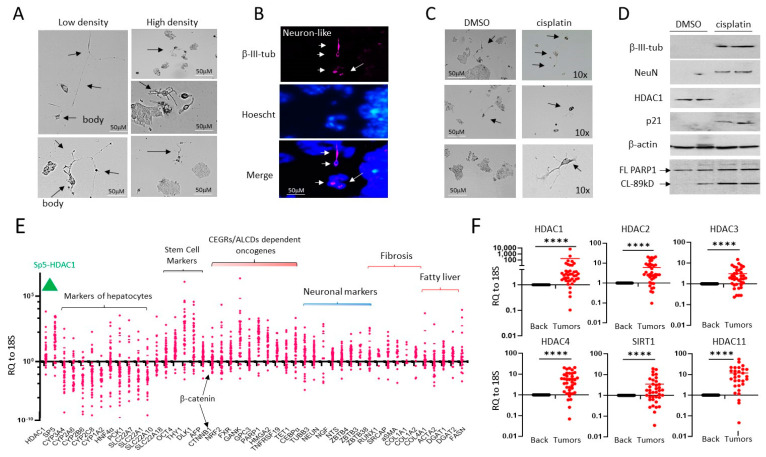
HepG2 cells contain a small portion of neuron-like cells that are resistant to cisplatin. (**A**) Images of HepG2 cells plated at high and low density. Arrows show cells that have the shape of neurons. (**B**) Immunostaining of HepG2 cells with β-III-tubulin. Arrows show positive cells. (**C**) HepG2 cells with the shape of neurons are resistant to cisplatin treatments. Arrows show cells that have the shape of neurons. (**D**) Western Blot analysis of proteins isolated from DMSO and cisplatin-treated HepG2 cells. Western blot of PARP1 shows that cisplatin-treated cells express PARP1 band (89 kD), a result of cleavage and an indicator of apoptosis. (**E**) mRNA expression of several pathways, including HDAC1-Sp5, hepatocyte markers, stem cells, and neuronal markers in a fresh biobank of HBL specimens (*n* = 42). RQ to18S shows ratios of mRNAs to the 18S RNA. (**F**) Examination of HDAC family in the biobank of pediatric liver cancers by QRT-PCR. A paired *t*-test was performed for background and tumor samples—**** denotes a *p*-value less than 0.0001. The whole gel images of western blots can be found in Appendix A.

**Figure 2 cancers-16-02300-f002:**
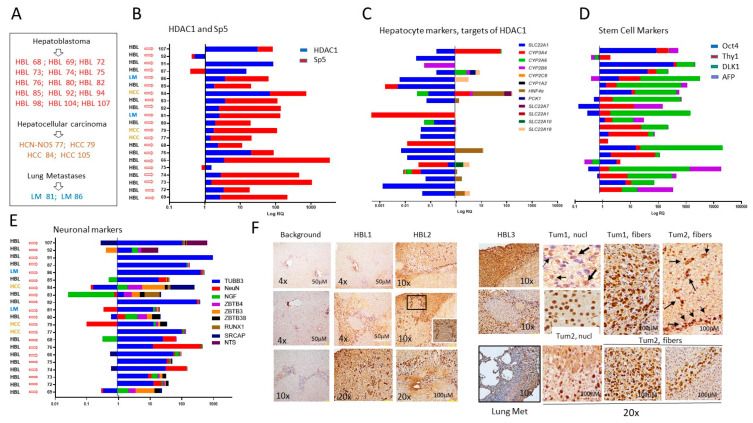
Tumor specimens of HBL and HCC patients have elevated HDAC1-Sp5 pathways and elevated stem cell markers and neuronal markers. (**A**) A list of HBL, HCC, and lung metastases patients, tumors of which were used for generating primary cultured cell lines. (**B**–**E**) QRT-PCR analysis of HDAC1 and Sp5 (**B**), hepatocyte markers (**C**), stem cell markers (**D**), and neuronal markers (**E**) in each patient. (**F**) Immunostaining of the original liver tumors and lung metastases with antibodies to HDAC1. 20× shows HDAC1-positive hepatocytes and fibers.

**Figure 3 cancers-16-02300-f003:**
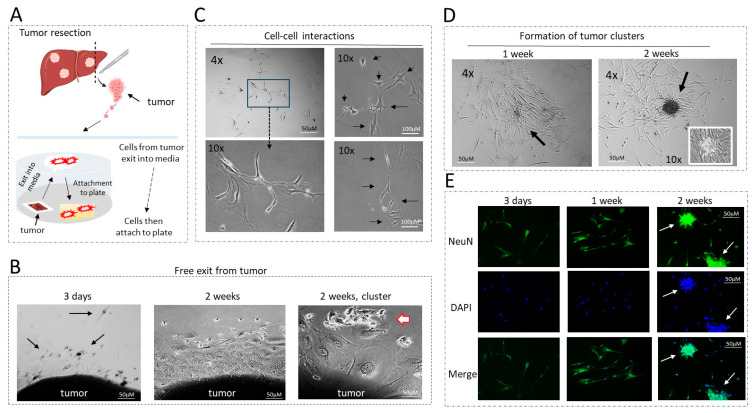
Generation and general description of patient-derived cells with metastatic activities. (**A**) A diagram showing the “cell free exit” protocol for generation of cultured cells. (**B**) Examples of the exit of cells from tumors at 3 days, 1 week and 2 weeks. Arrows show exiting cells with the shape of neurons. (**C**) Examples of cell–cell interactions. Images show physical interactions between 5 and 6 cells. Arrows show cells that interact with each other. (**D**) Examples of tumor cluster formation in one week and two weeks post-plating. Arrows point to the center of a cluster under 10× magnification. (**E**) NeuN staining of the hbl cell lines at different stages of tumor cluster formation. Arrows show NeuN-positive clusters.

**Figure 4 cancers-16-02300-f004:**
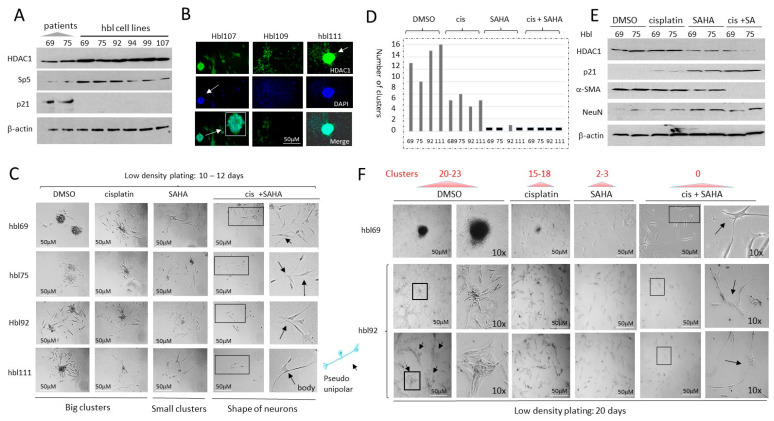
HDAC inhibition by SAHA increases cisplatin efficacy to block proliferation of hbl cells and formation of the tumor clusters. (**A**) Generated hbl cell lines have increased levels of the HDAC1-Sp5 pathway. Western Blot of protein extracts isolated from original liver tumors (patients) and from 6 generated hbl cell lines to HDAC1, Sp5 and p21. (**B**) Staining of three cell lines hbl107, hbl109, and hbl111 to HDAC1. Arrows show an HDAC1-positive tumor cluster. (**C**,**D**) Images of tumor clusters in hbl69, hbl75, hbl92 and hbl111 cells treated with DMSO, cisplatin, SAHA and cis+SAHA at 12 days after plating cells at low-density. (**C**) Representative cell images. Arrows show cells with the shape of neurons. (**D**) Number of clusters per plate. (**E**) Western blot analysis of HDAC1, p21, NeuN and α-SMA in treated hbl cells. (**F**) Images of tumor clusters in hbl69 and hbl92 cell lines at 20 days after initiation of the protocol. 10× images show tumor clusters and individual cells that have the shape of neurons (marked by arrows). The whole gel images of western blots can be found in Appendix A.

**Figure 5 cancers-16-02300-f005:**
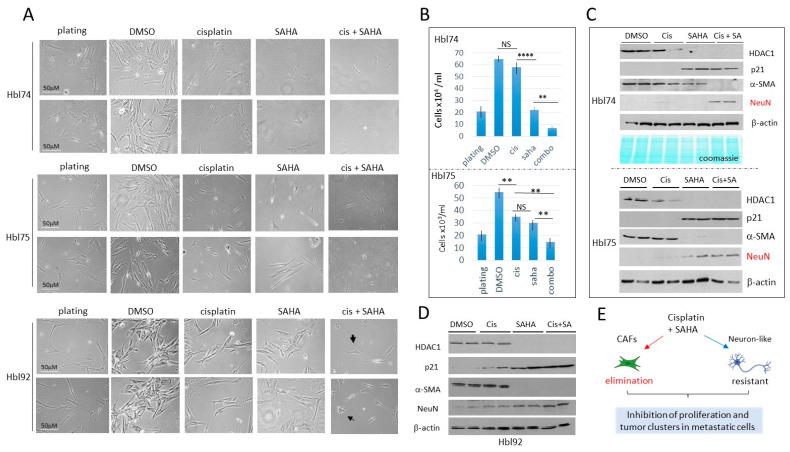
Combined treatments of hbl cells with cisplatin and SAHA eliminate CAFs, while neuron-like cells are not eliminated, but are growth-arrested. Cells were plated at high density and treated for 48 h. (**A**) Images of hbl74, hbl75 and hbl92 cells treated with DMSO, cisplatin, SAHA and cis+SAHA. Scale bars are identical for all images and are shown for the plated cells. (**B**) Counting of hbl74 and hbl75 cells after treatments. ** shows *p* < 0.01, **** shows *p* < 0.0001. NS—not significant. (**C**) Western Blot of proteins isolated from experimental plates of hbl74 and hbl75. (**D**) Western Blot of proteins isolated from experimental plates of hbl92. (**E**) A summary of studies of hbl cells treated with cisplatin and SAHA. The whole gel images of western blots can be found in Appendix A.

**Figure 6 cancers-16-02300-f006:**
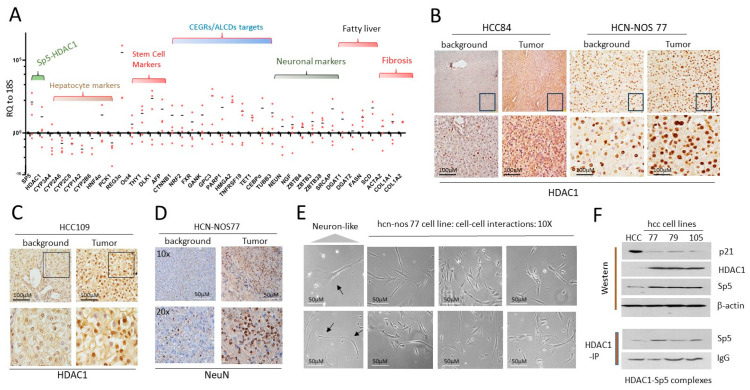
Generated hcc/hcn-nos cell lines maintained elevated HDAC1-Sp5 pathway observed in original liver tumors. (**A**) QRT-PCR analysis of HCC tumors from 4 patients. (**B**) Staining of the liver tumor of HCC84 and HCN-NOS77 patients with HDAC1. (**C**) Staining of HCC109 to HDAC1. (**D**) Staining of the liver tumor of HCN-NOS77 patient for NeuN. (**E**) Images of cells in hcn-nos77 cell line. The cells show intensive interactions with each other. Arrows show cells that have the shape of neurons. (**F**) Western Blot (**upper**) and Co-IP (HDAC1-IP, (**bottom**)) show that the generated cell lines preserved the increased HDAC1-Sp5 pathway. The whole gel images of western blots can be found in Appendix A.

**Figure 7 cancers-16-02300-f007:**
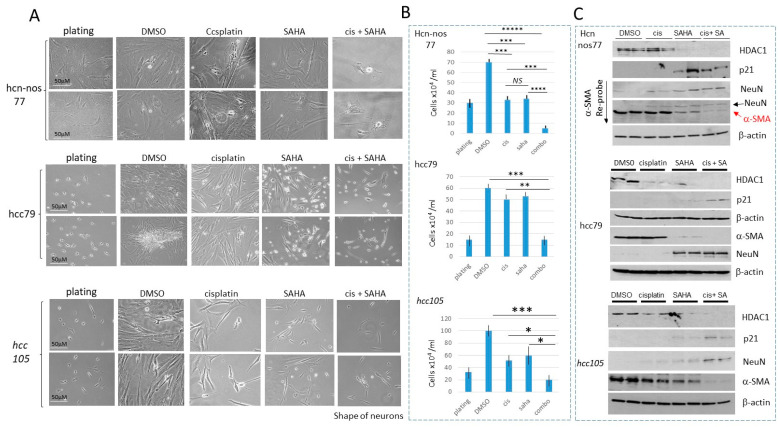
HDAC inhibition in hcc/hcn-nos cell lines increases cisplatin efficacy in eliminating cancer cells. (**A**) Images of hcn-nos77, hcc79 and hcc105 cells treated with DMSO, cisplatin, SAHA and cis+SAHA. Scale bars are identical for all images and are shown for the plated cells. (**B**) Calculations of the number of cells in each group. * shows *p* < 0.05, ** shows *p* < 0.01, *** shows *p* < 0.001, **** shows *p* < 0.0001, ***** shows *p* < 0.00001. NS—not significant. (**C**) Western blot analysis of HDAC1, p21, NeuN and α-SMA in each cell line. The whole gel images of western blots can be found in Appendix A.

**Figure 8 cancers-16-02300-f008:**
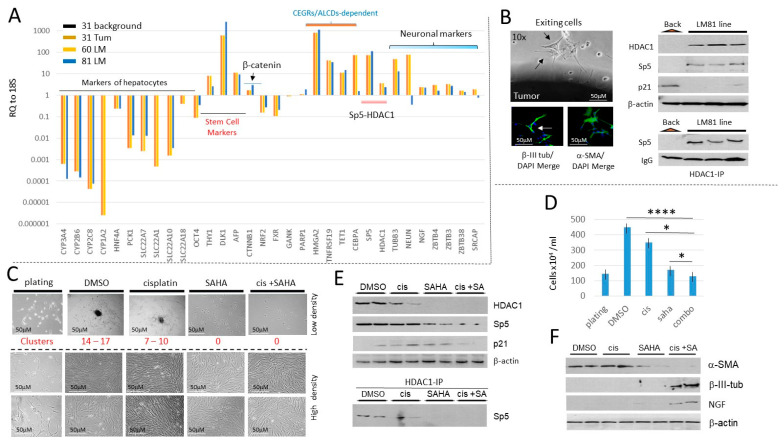
HDAC inhibition in a lung metastasis-derived cell line increases cisplatin efficacy to eliminate metastatic cells. (**A**) Characterizations of the cancer pathways in a primary liver tumor and in two lung metastases of a patient who had five total lung metastases. QRT-PCR analysis with a background region (adjacent to tumor), with liver primary tumor (#31), and with two lung metastases (#60 and #81) are shown. (**B**) Left: An example of exit of LM81 cells from the original Lung Metastasis and staining of the exiting cells with β-III-tubulin and α-SMA. Right, examination of HDAC1-Sp5 p21 pathway in the LM81 and in the LM 81cell line. (**C**) Treatments of LM81 cells with cisplatin, SAHA, and the combination of cis+SAHA. Red text shows the number of tumor clusters on the plates with low density plating. (**D**) The number of cells on plates with LM81 cells loaded at high density. * shows *p* < 0.05, **** shows *p* < 0.0001. (**E**) The HDAC1-Sp5 pathway is eliminated by combined treatments with cis+SAHA. The upper part shows levels of HDAC1, Sp5 and p21; the bottom shows HDAC1-IP and Western blot to Sp5. (**F**) Western blot analysis of proteins isolated from plates treated with DMSO, cisplatin, SAHA and cis+SAHA. The filters were probed for α-SMA, β-III-tubulin and NGF. The whole gel images of western blots can be found in Appendix A.

**Figure 9 cancers-16-02300-f009:**
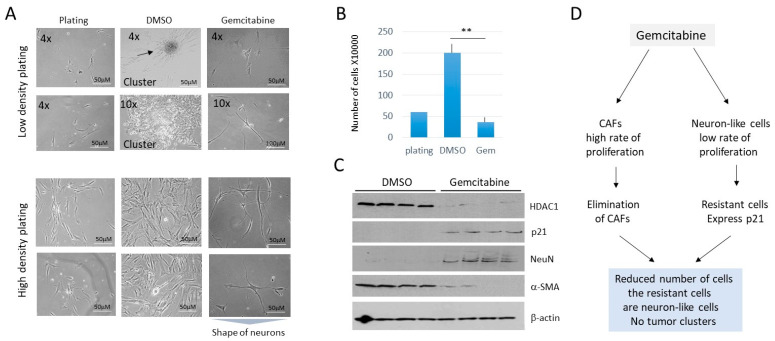
Treatment of LM81 cells with gemcitabine eliminates CAFs and inhibits the formation of tumor clusters and proliferation of cells. (**A**) Images of LM1 cells treated with DMSO and with 5 nM of gemcitabine. Cells were plated at low-density (**upper**) and high-density (**bottom**). (**B**) Bar graphs show calculations of high-density plated cells after treatments. ** shows *p* < 0.01. (**C**) A western blot was performed with proteins isolated from the 4 experimental plates for each treatment. (**D**) Summary of the treatments of LM1 cells with gemcitabine. The whole gel images of western blots can be found in Appendix A.

**Figure 10 cancers-16-02300-f010:**
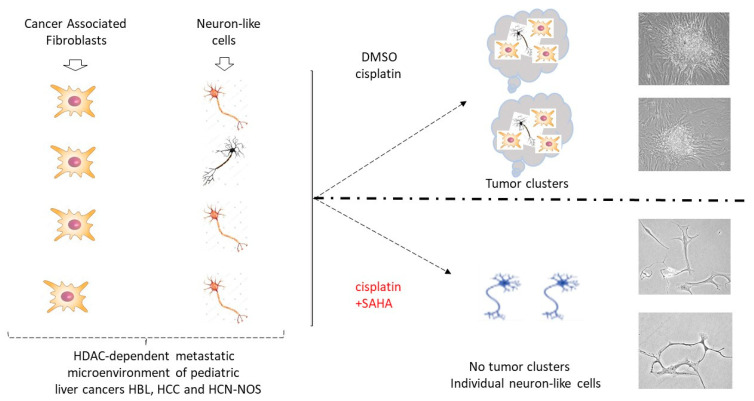
Inhibition of HDAC activity increases the ability of cisplatin to eliminate metastatic cancer cells and reduce the tumor clusters in pediatric liver cancers. The diagram shows a summary of the results presented in this manuscript. Right images show typical images observed on the experimental plates.

## Data Availability

The data presented in this study are available in this article and Appendix A.

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
