# Peer review of "Inhibition of Histone Deacetylase Activity Increases Cisplatin Efficacy to Eliminate Metastatic Cells in Pediatric Liver Cancers"

_cancers, 2024, doi:10.3390/cancers16132300_

Round 1

Reviewer 1 Report

Comments and Suggestions for Authors

This manuscript from the Timchenko group addresses the basis for acquired resistance to cis-platinum in hepatoblastoma.  Using both primary tumors, metastases and cell lines derived from them, they show that drug resistance is associated with the over-expression of HDAC and its associated transcription factor Sp5. Treatment of these with the HDAC inhibitor SAHA appears to restore sensitivity to cis-platinum and imparts neuronal-like properties to the surviving cells.

Major points

1. In Fig. 1, the authors claim that neuron-like cells are found during the cultivation of HepG2, that they are more abundant after culturing at low density and that they are resistant to cis-platinum.  However, no quantification is reported as to the fraction of these neuron-like cells that comprise this population under each set of growth conditions and the relative rate of killing of each group in response to cis-platinum. This issue persists through the remainder of the paper where only pictures if neuronal-like cells are shown without quantification.  Are the neuron-like cells resistant to cis-plat simply because they are not growing? This would seem to be the case, given that express p21.  Also, how permanent is this change? Can pure lines or at least colonies of neuron-like cells be obtained?

2. In Fig. 2, the authors provide a list of the cell “lines” they established. However, no explanation is provided as to the procedures that were used to generate these. Line 121 in the Materials and Methods section seems to explain this but the heading “Cell-free-exit protocol…” is confusing. Permanent, immortalized HB cell lines are notoriously difficult to generate (Rikhi RR, Spady KK, Hoffman RI, Bateman MS, Bateman M, Howard LE. Hepatoblastoma: A Need for Cell Lines and Tissue Banks to Develop Targeted Drug Therapies. Front Pediatr. 2016 Mar 21;4:22.).  Thus, a detailed explanation of how these lines were derived should be provided. Also, it is not clear whether these lines are truly immortalized or simply capable of limited in vitro proliferation for a relatively short time before senescing (like primary fibroblasts). Indeed, the authors refer to these “lines” throughout the manuscript as being “primary”. If indeed they do have limited replicative capacity, they should be more properly referred to as “strains”. This confusion should be addressed.

3. Fig. 4B, I assume that the figure depicts hbl174 cells at the top and hbl175 cells at the bottom but these should be labeled.  Also, the results showing there to be no differences between DMSO and cis-plt do not seem to agree with the images shown in panel A where the cis-plat. photos look much more like the “plating” images.

4. An obvious question is the origin of the CAFs and neuron-like cells being described in association with HBLs and HCCs. Most studies indicate that CAFs (at least in liver cancers) can have multiple and not necessarily mutually exclusive origins, including hepatic stellate cells, adipocytes, and macrophages (Yang D, Liu J, Qian H, Zhuang Q. Cancer-associated fibroblasts: from basic science to anticancer therapy. Exp Mol Med. 2023 Jul;55(7):1322-1332).  There is also considerable precedence for “transdifferentiation” with examples being the neuronal-type differentiation seen with treated neuroblastoma, and the formation of tumor-derived endothelial cells.  Although there is not much evidence for the tumor cell origin of CAFs or neuronal cells in HBL or HCC, I wonder whether this could change given that prior treatment with cis-platinum might induce differentiation.  The tumor cell origin of these cells could be addressed by asking whether they harbor any evidence for any of the driver mutations in such genes as b-catenin, hTERT or NRF2 that are typically associated with HBL.

5. It’s somewhat surprising to see that all the gene expression analysis performed in this paper relied on qRT-PCR.  Given that nearly 40 such markers were used, it’s hard to imagine that it would not have been faster, cheaper and more informative had RNAseq been done instead.  This would have had the extra benefit of allowing the authors to investigate neuronal and fibroblast pathways using much more unbiased approaches. It would also allow the authors to state with greater assurance the degree to which the cells resemble actual neurons or fibroblasts

6. One of the reasons why comment 5 is relevant is that, while the authors provide evidence that cis-platinum resistance is associated with high levels of HDAC expression and can be overcome with HDAC inhibitors, they provide no insights into the mechanisms by which this resistance is acquired or the pathways involved. One could speculate that it involves changes in the expression of a variety DNA damage/repair enzymes. An unbiased RNAseq experiment could confirm/refute this hypothesis while also potentially revealing other explanations.

7. SAHA (Vorinostat) is a pan-HDAC inhibitor as acknowledged by the authors in the Discussion. Therefore, caution must be exercised in attributing SAHAs restoration of cis-platinum sensitivity solely to its inhibition of HDAC1.  Instead, the authors should perform a knockdown of HDAC1 and show that the above effect is both reproducible and specific.  An elegant way to do this would be to use a regulatable shRNA vector and show that an initially cis-plat-resistant cell line that expressed high levels of HDAC1 became sensitive when shRNA was turned on and then again became resistant when it was turned off.  This could be monitored by following HDAC expression levels at each of these points.  Ideally, RNAseq could be performed to pinpoint the precise gene expression changes involved in these transitions. 

Minor points

1. Most readers should be familiar with the HDAC family but many may not be aware of the relationship with Sp5.  A few lines in the Introduction re the relationship of Sp5 to HDACs would help to improve understanding of this important axis before the reader is introduced to it in the Results section.

2. The Abstract begins by presenting the idea that CAFs and neuron-like cells may contribute to metastatic propensities (lines 27-28). It then immediately pivots to mentioning the examination of histone de-acetylase inhibitors as potential combinatorial therapy for HBs.  What’s the connection? The casual reader will want to know how the authors arrived at the hypothesis that HDAC inhibition might be of benefit.  

3. Line 55: b-catenin and NRF2 mutations in HBL are described as being “rare”.  Many investigators who have identified b-catenin mutations and NRF2 mutations or amplifications in >50% of human tumors would take issue with this.

4. Fig. 1E & F: in one panel, the y-axis is labeled “RG to 18S” and in the other it is labeled “RQ to 18S”.  Are these actually different and what term are they abbreviating? Also, in panel E, as well as throughout the entire paper, the level of significance attached to ****’s must be stated.  Finally, what is meant by the term “Back”?  I assume this means background (i.e. adjacent “normal” liver).  It should be labeled this way to better indicate the origin of these tissues.  Also, are they matched to the patients with the tumors?

5. Fig. 5A. y-axis needs to be labeled.

6. There are many poorly written and/or awkward sentences and overall, the manuscript could greatly benefit from a careful re-writing/editing

Examples (by no means complete):

Line 16

Lines 78-80

Lines 98-99

Lines 294-296

Lines 370-374. Here and elsewhere, the past-tense should be used to describe results.  In this paper, the authors switch back and forth. In some cases, this occurs within the same sentence!!! (see the entire paragraph: lines 414-430). Use past-tense in all cases to describe experiments that were done in the past

Comments on the Quality of English Language

See my minor comments to the authors.  The manuscript could definitely benefit from  a careful re-writing and making sure that experiments are all described in the past tense

Author Response

We would like to thank the reviewer for useful suggestions to improve the manuscript. The revisions are included in the text and marked by blue color.

Reviewer 2 Report

Comments and Suggestions for Authors

In this in vitro study, the authors showed that HDAC inhibition by SAHA enhances the antiproliferative effects of cisplatin and reduces tumor clusters in hepatoblastoma and hepatocellular carcinoma patient-derived cell models. They convincingly demonstrate that cisplatin alone does not lead to a reduction in cell number, whereas SAHA alone and in combination shows a very strong effect. The authors refer to the opposing protein and gene expression of relevant cell markers such as HDAC1 and the tumor suppressor protein p21. The expression of the protein Sp5 is also discussed. Additionally, the authors defined cell subgroups, such as neuron-like cells and cancer-associated fibroblasts (CAFs), identifying them as metastasis-initiating cell populations. They further demonstrated the synergistic benefits of HDAC inhibition and cisplatin treatment. The study impresses with its high proportion of primary cell material used and the analysis of the mentioned regulated proteins. Also, cells from lung metastases were used for treatment. However, there is a lack of precision in the description and interpretation in many places, making it difficult to follow the authors' arguments. The following points should be addressed to recommend the present manuscript for publication.

Major

1.     Materials and Methods Tumor Models – The authors state in the methods section that they used tumor material from 50 patients. However, the cell lines mentioned later refer to 20 cell lines. It is not clearly described how these cell lines were selected—the numbering is confusing. Additionally, information about the age of the patients would be helpful.

2.     The paper describes that the authors isolated cells with metastatic potential from the tumor tissue. What differentiates these cells from the primary tumor cells? Not every cell growing next to the tumor is automatically a metastatic tumor cell.

3.     In all images of the figures the Scale bar is missing. The images and cells appear differently magnified, making it unclear that cells in high density produce fewer neuron-like cells (Fig 1a).

4.     In general: Is there a control cell line that does not produce such described cell extensions at low density? In my opinion, every single tumor cell on a culture plate automatically form filopodia to establish contact with neighboring cells. Why should this be a characteristic of hepatocellular carcinomas?

5.     Fig 3a: It appears that p21 expression is lost with cell culture. While p21 expression is present in the patient bands, it is no longer detectable in the cell lines. Here, I would suggest a more cautious formulation of the sentence from line 246 onwards, as the reduced p21 expression is not necessarily automatic in both tumors and cell lines. The same applies to Fig 5F. It seems that primary tumors tend to exhibit p21 expression. When cells are transitioned into the growth phase of a cell line, HDAC1 expression increases again.

6.     3D und 3F: In both figures, a number of tumor clusters are indicated that do not match. How do the authors explain this discrepancy? What is the difference between clusters and individual cells? Wouldn't it also be interesting to determine the total cell count and compare it with the formation of individual clusters?

7.     Line 313: Is the combination of SAHA and Cisplatin truly synergistic or rather additive? A calculation regarding this would be helpful.

8.     L320: Here, the cancer-associated fibroblasts (CAFs) are mentioned for the first time, but it is not entirely clear where this cell population comes from. This needs to be described in more detail.

Minor

·        Line 119: typing error: Western Blot

·        Line: 132 ff: the explanation of high and low density is vague; it would be better to provide the cell number at the time of seeding.

·        Line: 148 ff: Antibodies – the dilutions used are missing.

·        Line 158ff: PCR – the RNA concentration used is missing

·        Line 164ff: Western blot – the protein concentrations used are missing.

·        1C - It is not entirely clear what the three images stacked vertically are supposed to show. Labels are missing. Are these images all from the same experiment?

·        It is not entirely clear what is meant by cisplatin-resistant cells. Are these cells that remain on the plate after treatment with cisplatin? However, these cells are not necessarily resistant.

·        Line 215: Here, RNA data is mentioned. Are there also protein data regarding the expression of HDACs? This would be very interesting, as it would demonstrate whether the cells also upregulate the protein itself and not just the genes.

Author Response

(The authors gave the same response as above.)

Round 2

Reviewer 1 Report

Comments and Suggestions for Authors

The authors have, for the most part, adequately addressed the points I raised in my critique.  However, they may have mis-understood Point #2.  My question was whether the HB tumor cell "lines" that were derived from primary tumors were immortalized or, if not whether, they eventual became senescent when maintained in vitro.  I DID refer back to the original Gulati paper suggested by the authors and, while it does discuss in some detail the "exit cell" protocol, I was unable to find any mention as to whether the actual tumor cells were immortalized.  This information must be included in the paper.

Author Response

Response: We agree with the reviewer, and we included additional information in the text regarding this issue.  Since all experiments in this manuscript were performed with fresh cell lines after 2-4 passages, at this stage we cannot say if the cells are immortalized. The following statement was included in the manuscript.  “We noted that all examined cell lines did not become senescent at these early passages. Future work will show if these cells might be immortalized and used for long term studies (lines135-137)”.

Reviewer 2 Report

Comments and Suggestions for Authors

I have major or minor criticisms and recommend the manuscript for publication in its present form.

Author Response

Response: We thank the reviewer for the useful suggestions what we have included in our previous version of the revised manuscript.